# Porous Thin-Wall Hollow $Co_3O_4$ Spheres for Supercapacitors with High Rate Capability

**Xiao Fan [1], Yongjiao Sun [2], Per Ohlckers [1],* and Xuyuan Chen [1],***

[1] Department of Microsystems, University of South-Eastern Norway, Campus Vestfold, Raveien 215, 3184 Borre, Norway; xiao.fan@usn.no

[2] Micro and Nano System Research Center, College of Information and Computer, Taiyuan University of Technology, Taiyuan 030024, China; yjsun88@126.com

* Correspondence: per.ohlckers@usn.no (P.O.); xuyuan.chen@usn.no (X.C.)

**Abstract:** In this study, a zeolitic imidazolate framework-67 (ZIF-67) was prepared as a precursor using a facile hydrothermal method. After a calcination reaction in the air, the as-prepared precursor was converted to porous thin-wall hollow $Co_3O_4$ with its original frame structure almost preserved. The physical and chemical characterizations of the nanomaterial were analyzed systemically. The electrochemical tests indicate that the obtained $Co_3O_4$ possesses large specific capacitances of 988 and 925 F/g at 1 and 20 A/g accompanying an outstanding rate capability (a 93.6% capacitance retention) and retains 96.6% of the specific capacitance, even after 6000 continuous charge/discharge cycles. These excellent properties mark the $Co_3O_4$ a promising electrode material for high performance supercapacitors.

**Keywords:** $Co_3O_4$; hollow structure; rate capability; supercapacitors

## 1. Introduction

Rapid technological development and accelerated natural resource consumption have largely increased the demand for efficient, environmentally-friendly, cost-effective, and safe energy storage devices [1–4]. In the last decade, supercapacitors—the new devices between conventional physical capacitors and lithium-ion batteries—have been extensively recognized as one of the most promising candidates for energy storage devices due to their high power density, long cycling lifespan, and fast charge/discharge process [5–11]. In general, supercapacitors can be divided into two categories according to their energy storage mechanism: One is the electric double-layer capacitor (EDLCs), which is mainly made of carbonaceous materials [12–16]; the other is the faradic redox reaction pseudocapacitor (PsCs), which usually utilizes transition metal oxides/hydroxides as electrode materials [17–21]. In particular, pseudocapacitors deliver much higher specific capacitance in comparison with electric double-layer capacitors, and receive considerable interest today [22,23].

Among various transition metal compounds, $Co_3O_4$ occupies a crucial position due to its superior theoretical specific capacitance (3560 F/g), nontoxicity, and low cost [24–26]. Meanwhile, the hollow structure stands out because of its novel interior geometry and surface functionality; this can possibly provide a large surface area and extra active sites, thus dramatically boosting the electrochemical properties [27,28]. To date, a great deal of efforts have been devoted to synthesizing $Co_3O_4$ with hollow morphologies such as hollow spheres [28], hollow nanocubes [29], and hollow cages [30]. Despite the great progress that has been made, the specific capacitance is still significantly below the theoretical value, especially at high current densities. Therefore, it is still a challenge to fully take advantage of this powerful hollow nanoscaled $Co_3O_4$ to achieve a large specific capacitance and a good rate capability.

In this work, we successfully prepared porous thin-wall hollow $Co_3O_4$ spheres from a zeolitic imidazolate framework-67 (ZIF-67) precursor through a simple and fast reaction. This nanostructure offers a large accessible surface area with numerous pathways. As a consequence, it exhibits a large specific capacitance of 988 F/g at 1 A/g with satisfactory cycling stability (96.6% retention after 6000 cycles). In particular, the specific capacitance of 925 F/g at even 20 A/g (slight decay of less than 7%) is extremely competitive among $Co_3O_4$-based electrode materials reported in literature. The pseudocapacitive performance manifests the great potential of the porous thin-wall hollow $Co_3O_4$ spheres as electrode materials for applications in supercapacitors.

## 2. Materials and Methods

### 2.1. Sample Preparation

All reagents were of analytical grade and were used as received without any further purification. First, 0.437 g of $Co(NO_3)_2 \cdot 6H_2O$ and 0.616 g of 2-methyl imidazole were briefly dissolved in respective 20 mL methanol solutions. Then, the latter solution was added dropwise into the prior one under vigorous stirring. After an ultrasonic bath for 20 min, the precipitates were collected by centrifugation and washed with methanol. Partial precipitates (0.02 g) were dispersed in a 15 mL methanol solution again. Subsequently, 0.175 g of $Co(NO_3)_2 \cdot 6H_2O$ was added into the separated solution. The resulting mixture was transferred to a Teflon-lined stainless-steel autoclave and heated at 120 °C for 1 h. The separated products were washed with methanol and dried at 60 °C. Finally, the porous thin-wall hollow $Co_3O_4$ spheres were obtained via heat treatment at 400 °C for 2 h.

### 2.2. Material Characterizations

The crystal structure of as-obtained $Co_3O_4$ was recorded by X-ray powder diffraction (XRD, EQUINOX 1000) with Cu–K$\alpha_1$ radiation ($\lambda$ = 1.5406 Å). Scanning electron microscopy (SEM, Hitachi SU8230) with energy-dispersive X-ray (EDX) spectrum was performed for characterizations of morphology and element composition. X-ray photoelectron spectroscopy (XPS) was collected on an ESCALAB 250Xi.

### 2.3. Electrochemical Measurements

The $Co_3O_4$, acetylene black, and polyvinylidene fluoride (PVDF) with a weight ratio of 80:15:5 were dispersed in n-methyl-2-pyrrolidone (NMP) solution and ground continuously for 10 min in a mortar to form a slurry. The working electrode was prepared by pressing the slurry on nickel foam (NF). The electrochemical measurements were carried out at room temperature in a typical three-electrode system (2M KOH as the electrolyte), in which a Pt net and Ag/AgCl were used as a counter electrode and a reference electrode, respectively. A cyclic voltammogram (CV) was conducted at various scan rates and galvanostatic charge/discharge (GCD) was tested at different current densities. The specific capacitance (*C*, F/g) based on CV is defined as:

$$C = \frac{\int i(V)dV}{2mv\Delta V},$$ (1)

and the specific capacitance based on GCD is given by:

$$C = \frac{It}{m\Delta V},$$ (2)

where $\int i(V)dV$ (V·A) is the integrated area of the CV curve, and *m* (g), *v* (V/s), $\Delta V$ (V), *I* (A/g), and *t* (s) are the mass of active material, scan rate, potential window, discharge current, and discharge time [31,32]. The electrochemical impedance spectrum (EIS) was determined over a frequency range

from 100 mHz to 100 kHz with an AC perturbation of 5 mV at open circuit potential. The cycling performance was evaluated through repetitive GCD tests.

## 3. Results and Discussion

The morphology of as-prepared $Co_3O_4$ was investigated by SEM. Figure 1 presents SEM images of the precursor (Figure 1a,b) and the calcined $Co_3O_4$ (Figure 1c,d). Surprisingly, the $Co_3O_4$ effectively inherited the morphology of the precursor (negligible size contraction and few broken pieces) and exhibits uniform spherical structures. Furthermore, the magnified image (Figure 1d) shows that the spheres tend to interconnect with one another and clearly confirms the hollow nature, as well as the thin shell thickness of the $Co_3O_4$ from the view of a broken piece. The diameter of the spheres lies between 500 and 600 nm. In addition, the shell is assembled from numerous nanoparticles, constructing a highly porous architecture. This structure results in more active sites on the surface and easier transportation of ions, leading to better electrochemical performance.

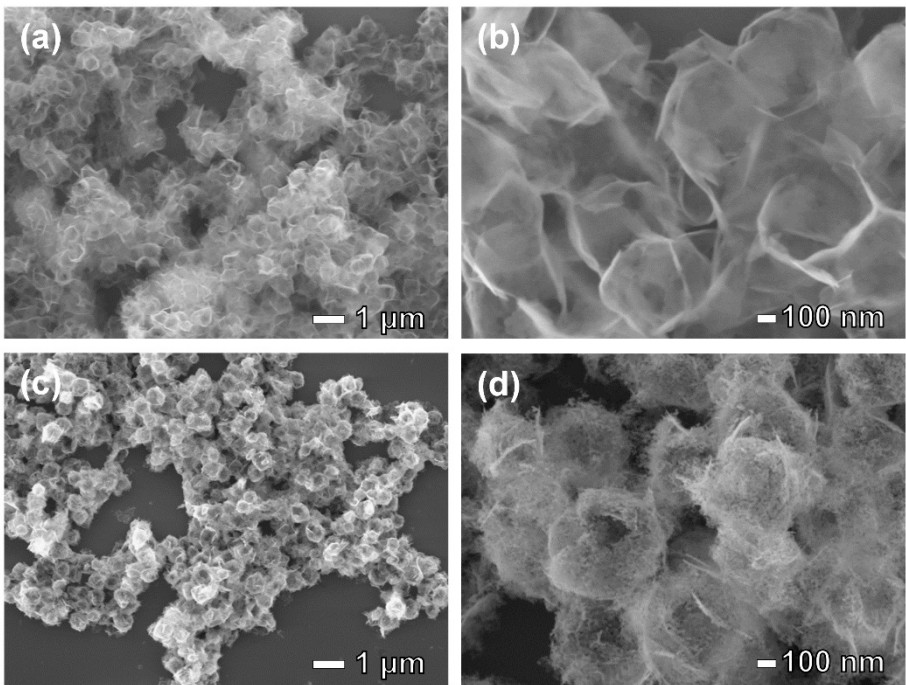

**Figure 1.** SEM images of (**a**) low- and (**b**) high-magnification of the precursor; (**c**) low- and (**d**) high-magnification of the calcined $Co_3O_4$.

The XRD patterns seen in Figure 2a, where the eight identified diffraction peaks located at 19°, 31.3, 36.9, 38.6, 44.9, 55.8, 59.5, and 65.4° are indexed to the (111), (220), (311), (222), (400), (422), (511), and (440) planes respectively, are perfectly consistent with the cubic phase of $Co_3O_4$ (JCPDS card No. 65-3103) [33]. The element compositions were confirmed by EDX, as depicted in Figure 2b. The peaks correspond to the elements Co and O, respectively. No other impurity element was observed. Both XRD and EDX characterizations reveal the high purity of the product.

The XPS was used to analyze the surface component and valence state of the as-obtained $Co_3O_4$. In the Co 2p core-level XPS spectrum seen in Figure 3a, the Co 2p was deconvoluted into two doublets. The peaks at 779.4 and 794.4 eV are related to $Co^{3+}$, and the peaks at 780.4 and 795.7 eV are assigned to $Co^{2+}$ [34]. In the high-resolution XPS spectrum of O 1s seen in Figure 3b, the O 1s can be fit into three different peaks at binding energies of 529.9, 530,9, and 532 eV, which are referenced to the lattice oxygen, the $OH^-$ species absorbed onto surface, and the multiplicity of physical and chemical absorbed water near surface, respectively [35]. This analysis of XPS further verifies the successful synthesis of the $Co_3O_4$.

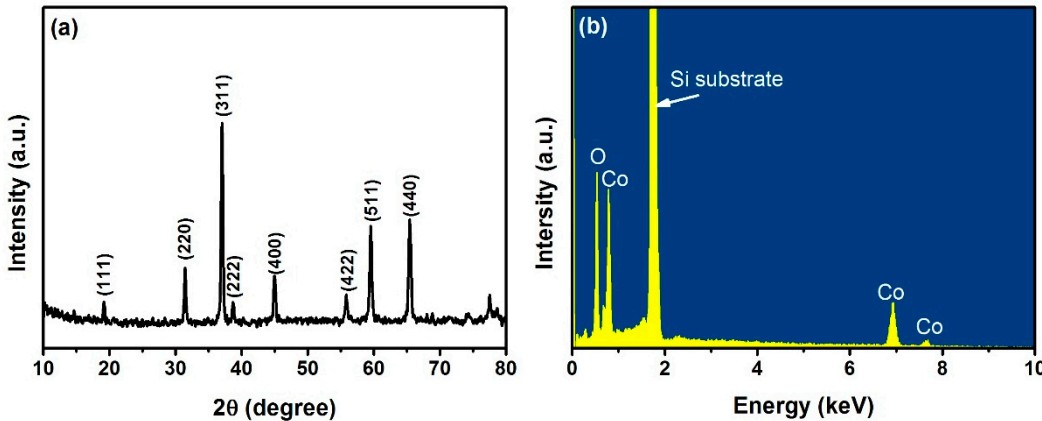

**Figure 2.** (**a**) XRD patterns and (**b**) energy-dispersive X-ray (EDX) spectrum of the obtained $Co_3O_4$.

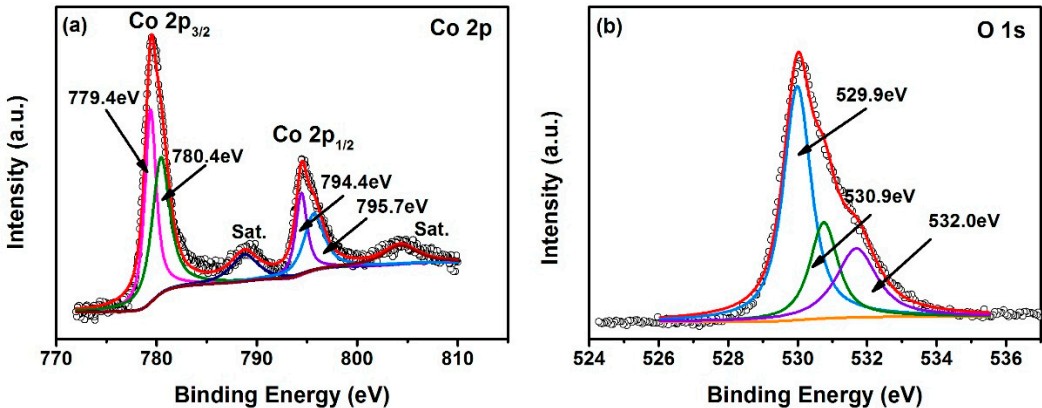

**Figure 3.** High-resolution X-ray photoelectron spectroscopy (XPS) scan of as-obtained $Co_3O_4$: (**a**) Co 2p region; (**b**) O 1s region.

The electrochemical performance of as-formed $Co_3O_4$ as an electrode material for supercapacitors was evaluated systematically. Figure 4a shows CV curves of the $Co_3O_4$ at various scan rates within a potential window of 0–0.55 V. As expected, the CV curve shape is totally different from those of electric double-layer capacitors (which have an almost rectangular shape), which reveals that the capacitance characteristics of the $Co_3O_4$ are those of typical pseudocapacitive capacitance. Additionally, two distinct pairs of redox peaks are visible in the CV curves. The first redox couple is ascribed to the faradic redox reaction of $Co^{2+}/Co^{3+}$, expressed as Equation (3) [29,30,36]:

$$Co_3O_4 + OH^- + H_2O \leftrightarrow 3CoOOH + e^-, \qquad (3)$$

and the second redox couple corresponds to the conversion of $Co^{3+}/Co^{4+}$, expressed as Equation (4) [29,30,36]:

$$CoOOH + OH^- \leftrightarrow CoO_2 + H_2O + e^-. \qquad (4)$$

When the scan rate increases from 5 to 100 mV/s, the peak current increases enormously. Furthermore, the anodic and cathodic peaks shift to higher and to lower potentials, suggesting classical pseudocapacitive behavior. The CV curve of the $Co_3O_4$ at 100 mV/s retains a redox shape similar to the original shape at 5 mV/s. It implies that the $Co_3O_4$ possesses an excellent rate capability, which is further confirmed by the calculated specific capacitances.

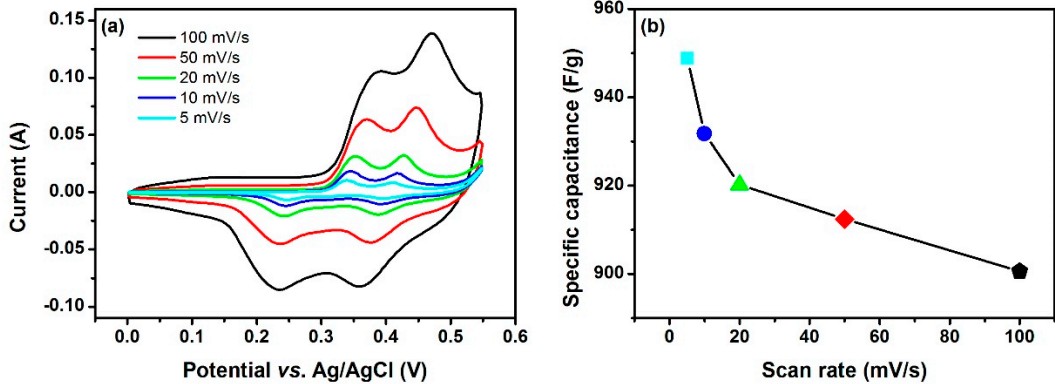

**Figure 4.** (**a**) Cyclic voltammogram (CV) curves and (**b**) specific capacitances of the $Co_3O_4$ electrode at various scan rates.

On the basis of Equation (1), the specific capacitances were determined to be 948.9, 931.8, 920.2, 912.4, and 900.6 F/g at scan rates of 5, 10, 20, 50, and 100 mV/s respectively, as shown in Figure 4b. The prominent specific capacitances most probably benefit from the porous hollow sphere nanostructure. The decrease in specific capacitances with an increasing scan rate can be understood in terms of the circuitous diffusion of $OH^-$ ions into the $Co_3O_4$. At a low scan rate, both the inner and outer surface of the material can be utilized for charge storage. At a high scan rate, the diffusion of $OH^-$ ions is more likely happen on the outer surface and only a small fraction of $OH^-$ ions can penetrate into the inner surface [37]. Nevertheless, the inevitable decrease of specific capacitances is not drastic, which demonstrates an excellent rate capability (maintaining almost 95% capacitance from 5 to 100 mV/s). This phenomenon illustrates that the $Co_3O_4$ is able to act as a "reservoir" of ions [38,39] to accommodate $OH^-$. The reservoir can guarantee the efficient proceeding of the faradic reactions at high scan rates (or high current densities) and in turn relieves the fading of the specific capacitances. Further evidence is provided by GCD measurement.

The discharge curves of the $Co_3O_4$ measured at different current densities in a voltage window of 0–0.55 V are presented in Figure 5a. In comparison with the discharge curves of electric double-layer capacitors (which are almost a straight line), two evident plateaus are well displayed. These coincide with the sequential redox reactions described by Equations (3) and (4), indicating the pseudocapacitive property of the $Co_3O_4$. According to Equation (2), specific capacitances of 988, 960, 947.9, 939.4, and 925 F/g can be delivered at current densities of 1, 2, 5, 10, and 20 A/g respectively, as depicted by Figure 5b. Due to the relatively insufficient active material involved in redox reactions under higher current densities, the boosting of the current densities results in the fading of specific capacitances. At a low current density, the $OH^-$ ions have adequate time to transfer at the interface between electrode and electrolyte than they do at a high current density [40]. It is worth noting that the rate performance of the $Co_3O_4$ is impressive (maintaining 925 F/g at 20 A/g, representing a 93.6% capacitance retention), which is significant for practical use of the material.

An EIS study is depicted in Figure 6. The first intersection point of the Nyquist plot with the real axis reflects the resistances ($R_s$), including the ionic resistance of the electrolyte, contact resistance at the interface of electrolyte and electrode, and intrinsic resistance of the material [39]. The $Co_3O_4$ modified electrode exhibits a very low $R_s$ of 0.52 Ohms. In the high frequency region, the charge-transfer resistance ($R_{ct}$) and double-layer capacitance ($C_{dl}$) correspond to the semicircle arc. The inset of Figure 6 shows that the semicircle has a small diameter, expressing a low charge-transfer resistance and a low interfacial resistance between the current collector and electroactive material [22,41]. In the low frequency range, the slope of the plot corresponds to the diffusive resistance of $OH^-$ ions (Warburg impedance, *W*) within the $Co_3O_4$. The plot presents an evident straight line with a slope greater than 45°, which indicates a valid electrolyte ion diffusion [30,34]. The equivalent circuit fitting the

impedance spectrum, which involves $R_s$, $R_{ct}$, $C_{dl}$, $W$, and pseudocapacitance ($C_{ps}$), is shown in the inset of Figure 6.

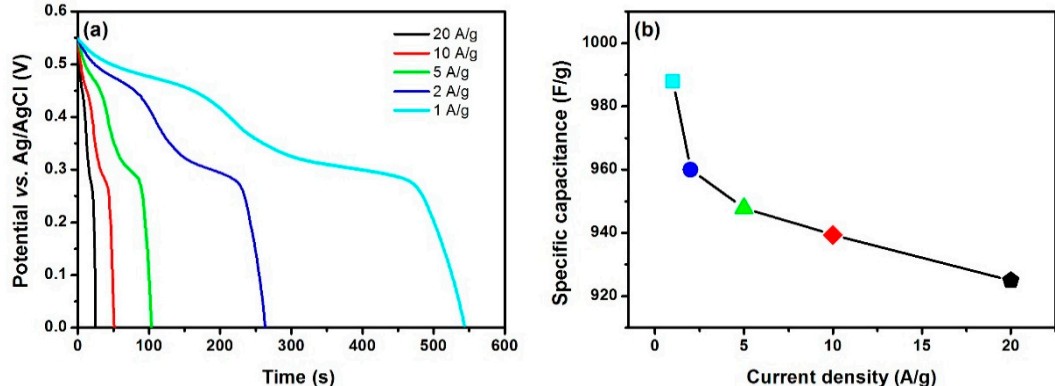

**Figure 5.** (**a**) Discharge curves and (**b**) specific capacitances of the $Co_3O_4$ electrode at different current densities.

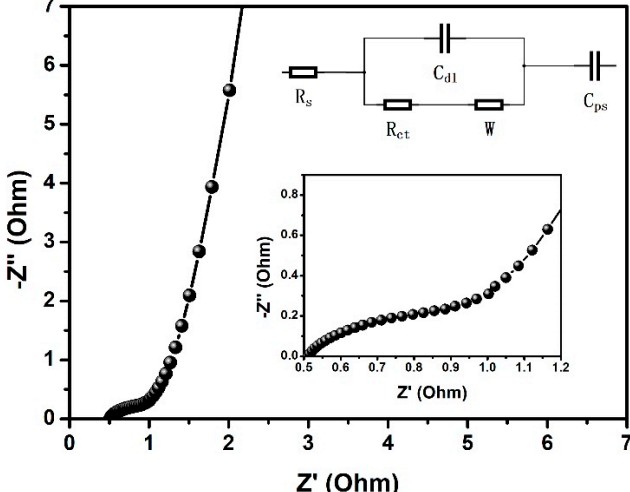

**Figure 6.** Electrochemical impedance spectrum (EIS) plot of the $Co_3O_4$ electrode over a frequency range from 100 mHz to 100 kHz. The insets show the equivalent circuit and an enlarged view of the high frequency region.

Long-term cycling stability is an important parameter required for practical application. Hence, the GCD tests were repeated for 6000 cycles at a current density of 20 A/g. As shown in Figure 7, 96.6% of the specific capacitance value is still retained. In addition, the coulombic efficiency is deduced from Equation (5) [37,39]:

$$\eta = \frac{t_d}{t_c}, \tag{5}$$

where $t_d$ (s) and $t_c$ (s) are, respectively, the discharge and charge times; a high efficiency exceeding 95% is achieved. This reveals that long-lasting cycling assessment does no obvious damage to the nanostructure and a feasible redox process is generated from the obtained material and verifies the remarkable electrochemical stability.

A comparison between our work and $Co_3O_4$-based electrode materials in current reports was made, as summarized in Table 1. Unfortunately, the high capacitive performances obtained through tremendous efforts usually accompany poor rate capability, which severely limits their wide application in high-power energy storage devices. The $Co_3O_4$ in this paper can meet the need of not only high specific capacitance but also excellent rate capability. In addition, the performance of the $Co_3O_4$ is

even better than some composites of cobalt oxides [34,42,43]. The superior electrochemical behaviors can first be explained by the stable morphology. Secondly, the large surface and inner space favors an efficient contact between active material and electrolyte, ensuring a high utilization rate of active material. Thirdly, the interconnected nanoparticles with an increased surface-to-bulk ratio offer more sites for ions to enter and, more importantly, allow facile ion diffusion at a high current density [44]. Finally, the hollow spheres are interconnected with one another, contributing to an even longer ionic diffusion channel [37].

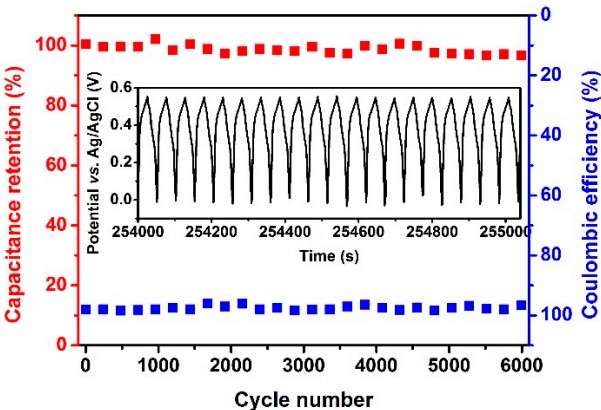

**Figure 7.** Cycling performance and coulombic efficiency of the $Co_3O_4$ electrode.

**Table 1.** Brief capacitance, rate, and cycling performance list of $Co_3O_4$-based electrode materials.

| Material | Specific Capacitance | Rate Capability | Stability-Cycles | Ref. |
|---|---|---|---|---|
| hollow $Co_3O_4$ spheres | 474.8 F/g at 1 A/g | 79% at 10 A/g | 99%-1000 | [37] |
| hollow $Co_3O_4$ spheres | 394.4 F/g at 2 A/g | 81% at 20 A/g | 92%-500 | [45] |
| hollow $Co_3O_4$ spheres | 342.1 F/g at 0.5 A/g | 69% at 10 A/g | 90%-2000 | [46] |
| hollow $Co_3O_4$ nanotubes | 1006 F/g at 1 A/g | 51% at 10 A/g | 91%-1000 | [29] |
| hollow $Co_3O_4$ cages | 948.9 F/g at 1 A/g | 56.6% at 40 A/g | – | [30] |
| hollow $Co_3O_4$ nanotubes | 404.9 F/g at 0.5 A/g | 87.8% at 20 A/g | 95%-2000 | [38] |
| hollow $Co_3O_4$ 3D-nanonet | 739 F/g at 1 A/g | 72% at 15 A/g | 90.2%-1000 | [39] |
| hollow $Co_3O_4$ flowers | 210 F/g at 0.5 A/g | 86% at 10 A/g | – | [46] |
| hollow $Co_3O_4$ corals | 527 F/g at 1 A/g | 78% at 10 A/g | 99%-5000 | [47] |
| hollow $Co_3O_4$ nanowires | 599 F/g at 2 A/g | 73% at 40 A/g | 91%-7500 | [48] |
| hollow $Co_3O_4$ boxes | 278 F/g at 0.5 A/g | 63% at 5 A/g | – | [49] |
| hollow $Co_3O_4$ dodecahedron | 1100 F/g at 1.25 A/g | 40% at 12.5 A/g | 95.1%-6000 | [50] |
| $Co_3O_4$ spheres | 837.7 F/g at 1 A/g | 93.6% at 10 A/g | 87%-2000 | [51] |
| $Co_3O_4$ spheres | 261.1 F/g at 0.5 A/g | 42% at 5 A/g | 90.2%-2000 | [52] |
| $Co_3O_4$ nanorods | 739 F/g at 5 mV/s | 52.5% at 100 mV/s | 103%-50000 | [53] |
| $Co_3O_4$ nanowires | 977 F/g at 2 A/g | 49.5% at 10 A/g | 90%-2000 | [54] |
| $Co_3O_4$ nanoflakes | 450 F/g at 1 A/g | 81% at 20 A/g | 92%-5000 | [55] |
| Mn doped $Co_3O_4$ | 668.4 F/g at 1 A/g | 62% at 10 A/g | 104%-10000 | [34] |
| Au decorated $Co_3O_4$ | 681 F/g at 0.5 A/g | 58% at 10 A/g | 83.1%-13000 | [42] |
| $CoO/Co_3O_4$ | 451 F/g at 1 A/g | 68.3% at 20 A/g | 108%-5000 | [43] |
| hollow $Co_3O_4$ spheres | 988 F/g at 1 A/g | 93.6% at 20 A/g | 96.6%-6000 | Ours |

## 4. Conclusions

In summary, porous thin-wall hollow $Co_3O_4$ spheres were successfully prepared through a simple approach and were characterized in detail. The electrochemical studies above unambiguously illuminate that the $Co_3O_4$ in this work shows prominent specific capacitance and cycling stability, which are attributed to its favorable structural features. Particularly, in comparison with other $Co_3O_4$-based electrode materials, it demonstrates an overwhelming rate capability (limited decay of 6.4% from

1 to 20 A/g), which is very critical for commercial applications. These results indicate the exciting supercapacitor application potential of our synthesized material. Moreover, the relationship between the structure and properties is worth investigating in depth in the future.

**Author Contributions:** X.F. investigated the research, performed the characterizations and analyzed the data; X.F. and Y.S. contributed equally to conduct the experiments and to write the manuscript; P.O. provided the critical feedback and designed the structure of the manuscript; P.O. and X.C. both equally supervised the experiments and refined the manuscript.

**Funding:** This research was funded by the Norwegian PhD Network on Nanotechnology for Microsystems, which is sponsored by the Research Council of Norway, Division for Science, under contract no. 221860/F60. X.F. was funded by China Scholarship Council.

**Acknowledgments:** Fruitful discussions with Einar Halvorsen and Pai Lu are acknowledged.

**Conflicts of Interest:** The authors declare no conflict of interest.

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
