# Peer review of "Porous Thin-Wall Hollow Co3O4 Spheres for Supercapacitors with High Rate Capability"

_applsci, doi:10.3390/app9214672_

Round 1
Reviewer 1 Report
Dear authors,
Your manuscript reveals a very nice body of work, I can see how much effort went into the measurements. The comparison table is very nice, I think you should improve it a bit so that it can be used by others. Suggestions for improvement are in attached pdf. My main suggestion would be to have the text be corrected by a proofreading service. It is not so expensive (maybe 150 Pounds), and will not cost you too much work. You efforts merit a well written paper!

Author Response
Response: Thanks for your thoughtful comments and suggestions. For the English expression problems you pointed out, we have checked point by point and revised all carefully. All changes made in the revised manuscript are clearly highlighted. You can find our responses to your questions below.
Question 1 (line 78): Could you provide the units you use (kg, V/s, V, A, s)?
Response: All units have been added in the revised manuscript.
Question 2 (line 90): ca is ambiguous. Perhaps better: lies between 300 and 700 nm or so…
Response: Thanks. The new sentence: “The diameter of the spheres lies between 500 and 600 nm”.
Question 3 (line 129): The number of significant numbers suggest an error of 50 ppm. Is that realistic?
Response: The specific capacitances were calculated by Equation (1). Considering that the order of magnitude of specific capacitances is close to 1000, the accuracy was set to 0.1.
Question 4 (line 150): “Due to the relatively insufficient active material involved in redox reaction under higher current densities” Is that a fact, or a hypothesis?
Response: It is a fact and well acknowledged in supercapacitors research field, which is proved by the reference in manuscript.
Question 5 (line 162 and 163): I suppose the series resistance relied on the electrode surface. How do I compare 0.52 Ohm with other publications? Would it make sense to give a resistance per area? I am not familiar with this method. Which arc are you addressing? Of the electrode? Than it is not explained in Methods. Perhaps describe electrode shape and area there?
Response: In EIS study, the Nyquist plot consists a small semicircle at the high frequency region and a straight line at the low frequency region, as shown in Figure 6. The inset shows enlarge view of the high frequency region, in which the semicircle arc can be observed clearly. The diameter of the semicircle reflects the charge-transfer resistance (Rct). In addition, the electrode shape (square) and area (1 cm × 1cm) have been described in the section of “Electrochemical Measurements”.
Question 6 (Table 1): This table is difficult to read. It helps to align numbers that you want your reader to compare at the decimal point. Might it be an idea to put this data in a plot, with on the vertical axis the rate and horizontally the capacitance? You could indicate the stability in the size of the symbol or so.
Response: As suggested by you, we have made careful modifications on the original table. We aligned the specific capacitances firstly. Further we only kept the rate capability (deleted the specific capacitances here) to make the researchers easy to read and compare. In addition, if we put the data in a plot, we fail to illustrate two key parameters (current density and cycling stability). We prefer to compare the performances of various Co3O4 electrode materials using a table rather than a plot. Thanks for your understanding.
Reviewer 2 Report
After carefully, reviewing the manuscript titled “Porous Thin-wall Hollow Co3O4 Spheres for Sueprcapacitors with High Rate Capability” it is advised to be accepted for published as is. In recent times, of nanoscale sciences and technology the study of nanomaterials has gained lot of interest to deal with fundamental limits of scaling and advancement. On the similar lines the study presented by authors here is of key importance especially in maintaining the high performance in supercapacitors. Study like the ones presented by author in this work add value to the scientific community and could be helpful for the general reference for the future work. In general, the manuscript is sound with scientific quality.
Author Response
Response: We do appreciate for your kindly approval and encouragement.
Reviewer 3 Report
The manuscript is written well and fits within the scope of this journal. I recommend this article for publication after minor revision. In this EIS results, it is important to describe the equivalent circuit considered for analysis. Please include it in the manuscript.
Author Response
Response: Thanks for your valuable comments. Based on your request, the equivalent circuit model fitting the obtained EIS result, which involves Rs, charge-transfer resistance (Rct), double-layer capacitance (Cdl), Warburg impedance (W) and pseudocapacitance (Cps), is shown in the inset of Figure 6.
See attached Word file for the figure.
Figure 6. EIS plot of the Co3O4 electrode over frequency range from 100 mHz to 100 kHz. The insets show the equivalent circuit and enlarged view of the high frequency region.

Reviewer 4 Report
In the manuscript "Porous Thin-wall Hollow Co3O4 Spheres for
Supercapacitors with High Rate Capability", Fan and co-authors prepared a new Co3O4 structure from a zeolite precursor, which have improved properties to be used as a supercapacitor.
The work is interesting and scientific meaningful. The results are clear and the methodology is described in a way that the experiments could be easily reproduces.
I recommend the paper for publication after minor adjustements. In particular, I would like to ask the authors if it is possible to include an atomic-scale understanding of the electrochemical and storage properties of this new materials, towards a rational desing of novel zeolite-like compounds.
Author Response
Response: Thanks for your thoughtful comments. We agree that the atomic-scale understanding of the electrochemical and storage properties of the obtained Co3O4 is benefic for the design of more novel pseudocapacitive materials. In our work, the desirable structures exhibit high specific capacitance, excellent rate capability and cycling stability. It can be attributed to the large surface and inner space, which favors the contact between active material and electrolyte [1]. Moreover, the highly porous architecture providing numerous pathways and more active sites results in easier and fast transportation of ions and electrons [2]. It is difficult for us to analyze the relationship between the structures and properties on atomic-scale so far. The idea will trigger our efforts for future studies in supercapacitors research field.
[1] Wang, Y.; Lei, Y.; Li, J.; Gu, L.; Yuan, H.; Xiao, D. Synthesis of 3D-nanonet hollow structured Co3O4 for high capacity supercapacitor. ACS Appl. Mater. Interfaces 2014, 6, 6739-6747.
[2] Wang, X.; Zhang, N.; Chen, X.; Liu, J.; Lu, F.; Chen, L.; Shao, G. Facile precursor conversion synthesis of hollow coral-shaped Co3O4 nanostructures for high-performance supercapacitors. Colloids Surf. A 2019, 570, 63-72.